# The Aerodynamics of the Curled Wake:
# A Simplified Model in View of Flow Control

Luis A. Martínez-Tossas, Jennifer Annoni, Paul A. Fleming, and Matthew J. Churchfield

National Renewable Energy Laboratory, Golden, CO USA

**Correspondence:** Luis A. Martínez-Tossas (luis.martinez@nrel.gov)

**Abstract.** When a wind turbine is yawed, the shape of the wake changes and a curled wake profile is generated. The curled wake has drawn a lot of interest because of its aerodynamic complexity and applicability to wind farm controls. The main mechanism for the creation of the curled wake has been identified in the literature as a collection of vortices that are shed from the rotor plane when the turbine is yawed. This work extends that idea by using aerodynamic concepts to develop a control-oriented model for the curled wake based on approximations to the Navier-Stokes equations. The model is tested and compared to time-averaged results from large-eddy simulations using actuator disk and line models. The model is able to capture the curling mechanism for a turbine under uniform inflow and in the case of a neutral atmospheric boundary layer. The model is then incorporated to the FLOw Redirection and Induction in Steady State framework and provides good agreement with power predictions for cases with two and three turbines in a row.

## 1 Introduction

A curled wake is a phenomenon observed in the wake of a wind turbine when the turbine is yawed relative to the free-stream velocity. When a wind turbine is yawed, the wake is not only deflected in a direction opposite to the yaw angle, but its shape changes. The mechanism behind this effect has drawn attention, not only from fluid dynamicists because of the interesting physics phenomena happening in the wake, but also from the wind turbine controls community who intends to use it to control wind farm flows (Fleming et al., 2017).

It has been shown that wake steering (redirection of the wake through yaw misalignment) can lead to an increase in power production of wind turbine arrays (Adaramola and Krogstad, 2011; Park et al., 2013; Gebraad et al., 2016). Previous studies have used large-eddy simulations (LES) and analytical tools to show the effects of yawing in the redirection of the wake (Jiménez et al., 2010; Bastankhah and Porté-Agel, 2016; Shapiro et al., 2018). Wind tunnel experiments have also been used to study the wake of a wind turbine in yaw (Medici and Alfredsson, 2006; Bartl et al., 2018). The curled wake mechanism, in the context of wind turbine wakes, was first identified by Howland et al. (2016) during a porous disk experiment, and in LES using an actuator disk model (ADM) and actuator line model (ALM). This mechanism was described by a pair of counter-rotating

vortices that are shed from the top and bottom of the rotor when the rotor is yawed. Further, these vortices move the wake to the side and create a curled wake shape. This mechanism was later confirmed and elaborated by the work of Bastankhah and Porté-Agel (2016) by performing experiments using particle image velocimetry of a scaled wind turbine and by doing a theoretical analysis using potential flow theory. Shapiro et al. (2018) show that the vorticity distribution shed from the rotor

because of yaw has an elliptic shape as opposed to only a pair of counter-rotating vortices. The curled wake has also been observed in LES with different atmospheric stabilities (Vollmer et al., 2016). Berdowski et al. (2018) were able to reproduce curled wake profiles by using a vortex method. Fleming et al. (2017) show that the generated vortices affect the performance of wake steering and motivate the development of engineering models (like the one in this paper), which include wake curling physics. Controllers based on such models would pursue different, and likely more effective, wind farm control strategies.

FLOw Redirection and Induction in Steady State (FLORIS) is a software framework used for wind plant performance optimization (Gebraad et al., 2016; Annoni et al., 2018). A wake model is used in FLORIS to compute the effect of wind turbine wakes on downstream turbines. Different models can be used inside of FLORIS to compute the wind turbine wakes, including the Jensen and Gaussian wake models (Bastankhah and Porté-Agel, 2016; Jensen, 1983). A review of control-oriented models can be found in Annoni et al. (2018). The model proposed in this work is a new wake model and it has been incorporated to

the FLORIS framework. The model is used to compute the wake from each turbine in a wind farm. In the curled wake model, the $V$ and $W$ velocity components generated by each turbine are superposed linearly. After computing the individual wakes, they are added using the sum of squares method (Katic et al., 1987).

In this work, we describe the aerodynamics of the curled wake, and propose a new, simple and computationally efficient model for wake deficit, based on a linearized version of the Navier-Stokes equations with approximations. The model is tested

and compared to LES using actuator disk and line models.

## 2   Aerodynamics of the Curled Wake: A Control-Oriented Model

Here, we develop a simplified model of the wake deficit considering the aerodynamics of the curled wake. We start by writing the Reynolds-averaged Navier-Stokes streamwise momentum equation for an incompressible flow:

$$u\frac{\partial u}{\partial x} + v\frac{\partial u}{\partial y} + w\frac{\partial u}{\partial z} + = -\frac{1}{\rho}\frac{\partial p}{\partial x} + \nu_{\text{eff}}\left(\frac{\partial^2 u}{\partial x^2} + \frac{\partial^2 u}{\partial y^2} + \frac{\partial^2 u}{\partial z^2}\right),\tag{1}$$

where $u$, $v$ and $w$ are the time-averaged velocity components in the streamwise, spanwise and wall-normal directions. $p$ is the time-averaged pressure, $\rho$ is the fluid density, and $\nu_{\text{eff}}$ is the effective (turbulent and molecular) viscosity. The flow field is now decomposed into a base solution and a perturbation about this solution. The base solution includes what we consider to be the main effects that convect the wake, including, but not limited to:

    1. The streamwise velocity profile

2. The rotational velocity from the shed vortices caused by yawing

    3. The rotational velocity due to the blade rotation.

The velocity components can be expanded as:

$$u = U + u', \qquad v = V + v', \qquad w = W + w',$$ (2)

where $U$, $V$, and $W$ are the streamwise, spanwise, and wall-normal velocity components from the base solution, and $u'$, $v'$, and $w'$ are the perturbation velocities. $U$, $V$, and $W$ represent a base flow that is responsible for convecting the wakes. And $u'$, $v'$, and $w'$ are the perturbation velocities about the base solution which represent the wake deficits. We use a reference frame where $x$ is the streamwise direction, $y$ is the spanwise direction, and $z$ is the wall-normal direction. Assuming that the inflow is fully developed, the base solution is only a function of the spanwise and wall normal directions.

Linearizing the Euler momentum equation for the streamwise component leads to:

$$U\frac{\partial u'}{\partial x} + V\frac{\partial (U + u')}{\partial y} + W\frac{\partial (U + u')}{\partial z} = -\frac{1}{\rho}\frac{\partial p}{\partial x} + \nu_{\text{eff}}\left(\frac{\partial^2 u'}{\partial x^2} + \frac{\partial^2 U + u'}{\partial y^2} + \frac{\partial^2 U + u'}{\partial z^2}\right).$$ (3)

For this initial test of the model, the pressure gradient is assumed to be zero, which is a valid approximation in the far wake but not necessarily near the rotor. Also, we assume that the streamwise velocity, $U$, is not a function of the streamwise ($x$) and spanwise ($y$) coordinates. We also assume that the viscous force from the boundary layer is balanced by its pressure gradient. With these simplifications, we are left with the equation:

$$U\frac{\partial u'}{\partial x} + V\frac{\partial u'}{\partial y} + W\frac{\partial (U + u')}{\partial z} = \nu_{\text{eff}}\left(\frac{\partial^2 u'}{\partial x^2} + \frac{\partial^2 u'}{\partial y^2} + \frac{\partial^2 u'}{\partial z^2}\right).$$ (4)

Equation 4 describes the downstream evolution of the wake deficit, $u'$. We note that the only velocity component being solved is the streamwise component, $u'$. The base solution for the flow $(U, V, W)$ includes the spanwise effects from different features, such as rotation and curl. Because these effects are the main drivers for wake deformation, the spanwise perturbations, $v'$ and $w'$, are assumed to be zero. This assumption reduces the model to a single partial differential equation. Equation 4 will be solved numerically in the next section. The model does not use the continuity equation so mass conservation is not strictly enforced for the perturbation velocities.

## 2.1 Curled Wake

The curled wake effect is added to the model by adding a distribution of counter-rotating vortices to the base flow solution. Figure 1 shows a schematic of the rotor and a collection of vortices being shed at the rotor plane. The superposition of all the vortices leads to a spanwise velocity distribution that creates the curled wake shape. Each vortex is described as a Lamb-Oseen vortex with a tangential velocity distribution given by:

$$u_t = \frac{\Gamma}{2\pi r}\left(1 - \exp\left(-r^2/\sigma^2\right)\right),$$ (5)

where $u_t$ is the tangential component of the velocity, $r$ is the distance from the vortex core, $\Gamma$ is the circulation strength, and $\sigma$ determines the width of the vortex core. The circulation related to the strength of each vortex is still unknown. We assume that the problem is symmetric and that all the circulation leaves the disk through the shed vortices. In the current implementation,

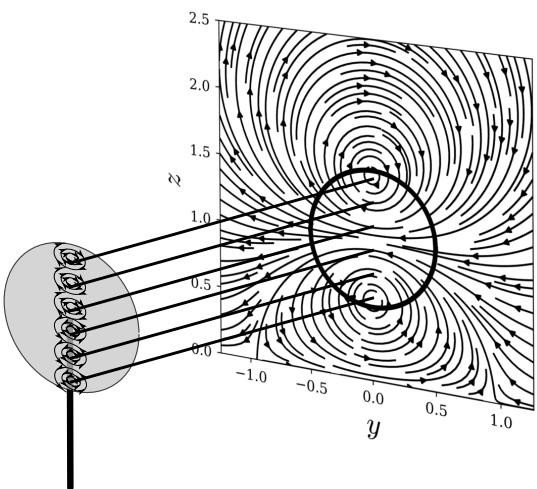

**Figure 1.** Diagram showing a collection of vortices shed from the rotor plane with the corresponding downstream distribution of spanwise velocities due to the superposition of the vortices.

the vortices are assumed not to decay as they are convected downstream; however, it would be possible to incorporate this effect into the model. It is important to note that the vorticity shed has an elliptic distribution according to the shape of the rotor (Shapiro et al., 2018). The total circulation is then: $\Gamma = \rho D U_\infty F_L$, where $\rho$ is the fluid density, $D$ is the rotor diameter, $U_\infty$ is the inflow velocity at hub height, and $F_L$ is the force perpendicular to the inflow wind. The force component perpendicular to

the inflow wind generates circulation. If we use the definition of the thrust coefficient, it is now possible to redefine the strength of the circulation as a function of thrust coefficient and yaw angle (Shapiro et al., 2018) as:

$$\Gamma = \frac{\pi}{8} \rho D U_\infty C_T \sin\gamma \cos^2\gamma, \tag{6}$$

where $C_T$ is the thrust coefficient for the given inflow velocity, and $\gamma$ is the yaw angle. The discrete elliptic distribution of shed vortices added to match the shape of the disk is:

$$V = \sum_{i=1}^{N} \frac{y_i \Gamma_i}{2\pi \left(y_i^2 + z_i^2\right)} \left(1 - \exp\left(-(y_i^2 + z_i^2)/\sigma^2\right)\right), \tag{7}$$

$$W = \sum_{i=1}^{N} \frac{z_i \Gamma_i}{2\pi \left(y_i^2 + z_i^2\right)} \left(1 - \exp\left(-(y_i^2 + z_i^2)/\sigma^2\right)\right), \tag{8}$$

where i is the index denoting each of the vortices distributed on a line between the top and bottom of the rotor diameter, $N$ is the total number of vortices, and the coordinates $y_i$ and $z_i$ are centered at the location of the shed vortex. The size of the vortex

core is set to $\sigma = D/5$. The choice of $\sigma$ intends to represent a realistic vortex core size for an elliptic distribution. This value represents $\sigma/D \sim 0.2$, which is on the order of the optimal size for flow over an airfoil (Martínez-Tossas et al., 2017). It is

possible to use other values, however, in the simulations presented, this value is a good compromise between a physical width and numerical stability. The strength of each vortex is associated with the elliptical distribution by:

$$\Gamma_{i} = -4\Gamma_0 \frac{r_{i}^2}{N D^2 \sqrt{1 - (2r_i/D)^2}}, \tag{9}$$

where $r_i$ is the radial location of the i-th vortex in the rotor coordinate system and $\Gamma_0 = 4/\pi\,\Gamma$ is used to ensure that the total

amount of circulation $\Gamma$ is conserved. This means that each vortex has a unique amount of circulation. When the distribution of circulation in Equation 9 is integrated, the total circulation is obtained. The results use $N =200$ discrete vortices, which has shown that this has little effect on the solutions, and values around $N = 20$ provide the same results. Noted that, when using just the two tip vortices in the top and bottom of the rotor, the shape of the curled wake does not match the simulation results. For this reason, an elliptic distribution of shed vorticity, which matches the shape of the rotor, should be used.

## 2.2   Wake Rotation

It is important to include wake rotation in the model, because the rotation will move the wake in a preferred direction. Wake rotation is taken into account by adding a tangential velocity distribution that is caused by the rotation inside the rotor area. The tangential induction factor is defined as:

$$a' = \frac{(a - a^2)\,R^2}{r^2\,\lambda^2} \tag{10}$$

where $a$ is the induction factor based on the thrust coefficient from standard actuator disk theory, $R$ is the rotor radius, $r$ is the radial distance from the center of the rotor, and $\lambda$ is the tip speed ratio (Burton et al., 2002). From this equation, the tangential component of the velocity $u_t$ is a singular vortex with $1/r$ behavior:

$$u_t = 2a'\lambda U_\infty r/R = \frac{2(a - a^2)U_\infty R}{\lambda r}. \tag{11}$$

A Lamb-Oseen vortex is used to desingularize the behavior near the center of the rotor. The circulation strength for the wake

rotation vortex based on Equation 10 is now:

$$\Gamma_{\mathrm{wr}} = 2\pi\,(a - a^2)\,U_\infty\,D/\lambda. \tag{12}$$

We assume that the wake rotation vortex does not decay or deform as it moves downstream. This is not necessarily true, as turbulence mixing will decrease the wake rotation. However, the present model does not diffuse the spanwise velocity components and some of the errors in the model can be attributed to this. The current implementation uses a Lamb-Oseen

vortex with a core size $\sigma = D/5$ to eliminate numerical instabilities caused by high velocities near the vortex core, but other values could be used.

## 2.3 Atmospheric Boundary Layer

The atmospheric boundary layer can be specified as part of the background flow. A profile including streamwise and spanwise velocity components can be specified. The streamwise profile is described by using a power law:

$$U = U_h \left( \frac{z}{z_h} \right)^{\alpha}, \tag{13}$$

where $U_h$ is the velocity at hub height, $z_h$ is the hub height, and $\alpha$ is the shear exponent. Also, it is possible to add a profile for the spanwise velocity component to take veer into account. In order to avoid numerical instabilities, the minimum wind speed is set to 20% of $U_h$. This happens close to the bottom wall where the results do not affect the solution in the wake.

## 2.4 Turbulence Modeling

The turbulent viscosity in Equation 4 is determined by using a mixing length model. The turbulent viscosity in the atmospheric boundary layer is dependent on a mixing length, $\ell_m$, and a velocity gradient (Pope, 2001). The mixing length for flows in the atmospheric boundary layer is defined by:

$$\ell_m = \kappa z \frac{1}{(1 + \kappa z / \lambda)} \tag{14}$$

where $\kappa$ is the von-Kàrmàn constant, $z$ is the distance from the wall, and $\lambda = 15\mathrm{m}$ is the value reached by $\ell_m$ in the free atmosphere (Blackadar, 1962; Sun, 2011). Now the turbulent viscosity is given by:

$$\nu_T = \ell_m^2 \left| \frac{du}{dz} \right|, \tag{15}$$

where $\frac{du}{dz}$ is the streamwise velocity gradient in the wall-normal direction.

## 2.5 Ground Effect

The presence of the ground will have an effect on the shed vortices. The ground effect is incorporated by applying a symmetry boundary condition at the ground (Bastankhah and Porté-Agel, 2016). This is done by using Equations 7 and 8, with the $y$- and $z$-coordinates placed below the ground and inverting the sign of the circulation. This condition has a more dominant effect on the vortices close to the ground as they interact with the boundary.

## 2.6 Superposition of Solutions

Superposing all the effects mentioned earlier leads to a base flow that includes all the features presented. The linearized equation allows us to add features by superposing them onto the velocity components. Notice that in this implementation of the model, the base solutions is a function of only the spanwise directions, $y$ and $z$, and there is no dependency on the streamwise coordinate, $x$. However, dependency on the streamwise direction could also be included in the base solution. The vortices do induce motion on each other and the spanwise component of the momentum equation should be used. However, these motions are smaller than the streamwise motions and solving one equation as opposed to three reduces computational cost significantly.

## 2.7 Initial and Boundary Conditions

The initial condition for the perturbation velocity, $u'$, is specified as the yawed disk projected onto a plane normal to the streamwise direction. This shape represents the shape of the wake downstream right after the rotor. The exact shape of the wake near the rotor is much more complicated, and is not taken into account in the present model. The initial profile is set as a uniform distribution of wake deficit ($u' = -2 a U_\infty$) inside the rotor projected area, where $a$ is the induction. This step function is smoothened using a Gaussian filter to avoid numerical oscillations in the spanwise directions. The lateral boundaries are set to zero perturbation ($u' = 0$) because there is no wake in that region.

## 3 Numerical Solution

It is now possible to discretize Equation 4 and solve it numerically. Because the time derivative and pressure-gradient terms were dropped, the equation is parabolic, and it can be solved as a marching problem. The equation is solved by starting from an initial condition at the rotor plane and marching downstream. This is done by using a first-order upwind discretization for the streamwise derivative and second-order finite differencing for the spanwise derivatives. The discrete equation is now:

$$u'_{[i+1,j,k]} = u'_{[i,j,k]} - \frac{\Delta x}{(U+u')_{[i,j,k]}} \left( W_{[i,j,k]} \frac{(U+u')_{[i,j,k+1]} - (U+u')_{[i,j,k-1]}}{\Delta z} + \right.$$
$$\left. V_{[i,j,k]} \frac{u'_{[i,j+1,k]} - u'_{[i,j-1,k]}}{\Delta y} - \nu_{\text{eff}} \nabla^2 u'_{[i,j,k]} \right), \tag{16}$$

where $i, j$, and $k$ are the indices denoting the grid points in the $x$, $y$, and $z$ coordinates, $\nabla^2$ are the wall-normal and spanwise components in the Laplacian operator and $\nu_{\text{eff}}$ is the effective viscosity. Because of the explicit form of the equation, it is possible to include the non-linear term $u' \frac{\partial u'}{\partial x}$, which is not present in Equation 4. We note that this term has a small effect on the solution, and not including it provides similar results. The resolution used is on the order of 30-40 grid points per diameter. The typical domain size needed to capture the wake is on the order of $3D$ in the spanwise and wall-normal directions, and as long as needed for the downstream location. The computational expense of the algorithm without any optimization is small ($\sim$1-3 s) and can be used to generate curled wake profiles quickly.

### 3.1 Numerical Stability

The proposed numerical method uses a forward-time, centered-space method (Hoffman and Frankel, 2001). This algorithm is explicit, meaning that it can become unstable for certain conditions. Using Equation 16, we can establish the numerical criteria for stability from the forward-time, centered-space method (Hoffman and Frankel, 2001). Equation 17 shows a guideline for the stability requirement of the algorithm:

$$\left( \frac{\Delta x}{\Delta y} \right)^2 \left( \frac{W}{U} \right)^2 \leq 2 \frac{\Delta x}{\Delta y^2} \frac{\nu_{\text{eff}}}{U} \leq 1 \tag{17}$$

This stability criterion is based on a two-dimensional equation and the equation we are solving is three dimensional. However, after testing various conditions, this criterion served as a good guideline for the three-dimensional version of the equation.

After some algebraic manipulation, it is possible to show that the maximum grid spacing in the streamwise direction, $\Delta x$, is independent of the spanwise grid resolution. Equation 17 can be written as:

$$\Delta x \leq 2\nu_{\text{eff}} \frac{U}{W^2}, \qquad \Delta y \geq \sqrt{\frac{2\nu_{\text{eff}}\Delta x}{U}} \tag{18}$$

After testing the model with several grid spacings, it was found that a resolution on the order of $D/\Delta \sim 30 - 40$ provided converged results for the model without numerical oscillations. In the case of laminar inflow, an arbitrary viscosity needs to be added to stabilize the numerical solution. After experimenting with different values, an effective viscosity based on a Reynolds number

$$Re = \frac{UD}{\nu} \approx 10^4 \tag{19}$$

proved to be sufficient to stabilize the solution.

## 4   Comparison Between the Model and Large-Eddy Simulations

In this section, we compare the proposed model to LES with an actuator disk/line model. Different simulations are used to test the proposed model: 1) a simulation of an isolated turbine using the ADM under uniform inflow, 2) a simulation of an isolated turbine using the ALM under uniform inflow, and 3) a simulation of a turbine using the ALM inside the atmospheric boundary layer under neutral stability conditions.

### 4.1   Actuator Disk/Line Model under Uniform Inflow

Here, we compare the results from the model to LES of a wind turbine under uniform inflow of a turbine using an actuator disk/line model under uniform inflow from Howland et al. (2016). The yaw angle for these simulations is $\gamma = 30^o$. First, the model is compared to a simulation using an actuator disk without rotation. Figure 2 shows downstream planes with contours of streamwise velocity normalized by the inflow velocity, for the case of an ADM without rotation (thrust only). The streamlines are based on the cross-stream components of velocity. The overall shape of the curled wake is well-captured by the model. The overall streamline shapes between the LES and proposed wake model are similar. The pair of counter-rotating vortices are clearly visible in both cases; however, the LES computes streamlines with a more complex shape than the simpler model is capable of capturing. The resulting effect is that, in both cases, the wake deficit cross sections are deformed and curled in a similar fashion. The model does not contain the tower, which is present in the simulation; however, this has a small effect on the wake.

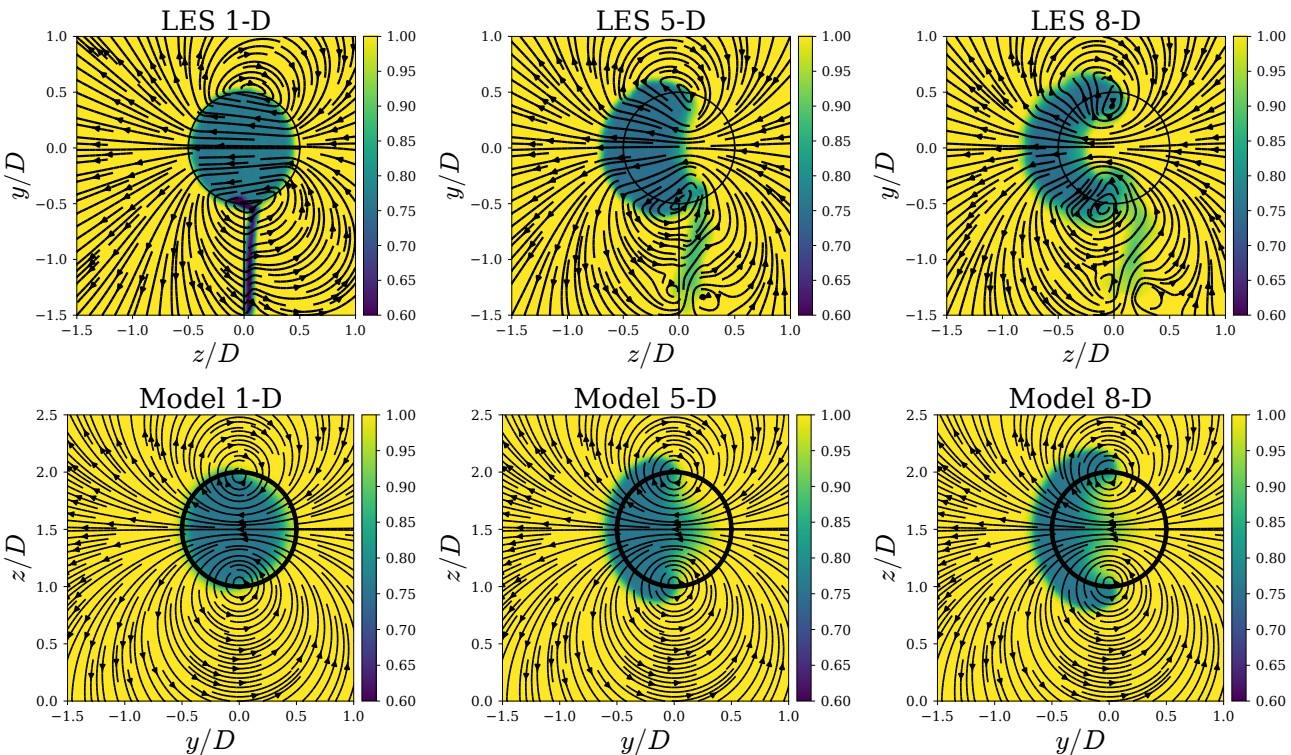

**Figure 2.** Comparison of streamwise velocity contours between a large-eddy simulation (LES) using an actuator disk model without rotation under uniform inflow from Howland et al. (2016) (top) and the proposed model (bottom). The streamlines show the spanwise velocity components.

Figure 3 shows downstream velocity contours for the case of an LES using an actuator line model under uniform inflow (Howland et al., 2016) and the proposed model including curl and rotation. The difference between the model implementation in this case, and the case of the actuator disk model is that rotation of the wake has been added. Again, the streamlines are similar in both cases, but the LES produces more complex patterns. Further, the resultant wake deficit deformation is also
5    similar in both cases, with more deficit remaining at the top of the wake. In this case, asymmetry is observed with respect to the centerline across the $y$-axis. This asymmetry is caused by the wake rotation induced by the torque applied to the fluid by the rotor. Interestingly, the combination of curl and rotation pushes most of the deficit in a preferred direction. In this case, it is pushed upward in the positive $z$ and negative $y$ directions. This insight can be used to steer the wake accordingly. We also observe that in the LES, the wake diffuses faster than in the model. This is because the model does not take yet into account
10   the turbulence generated by the wake.

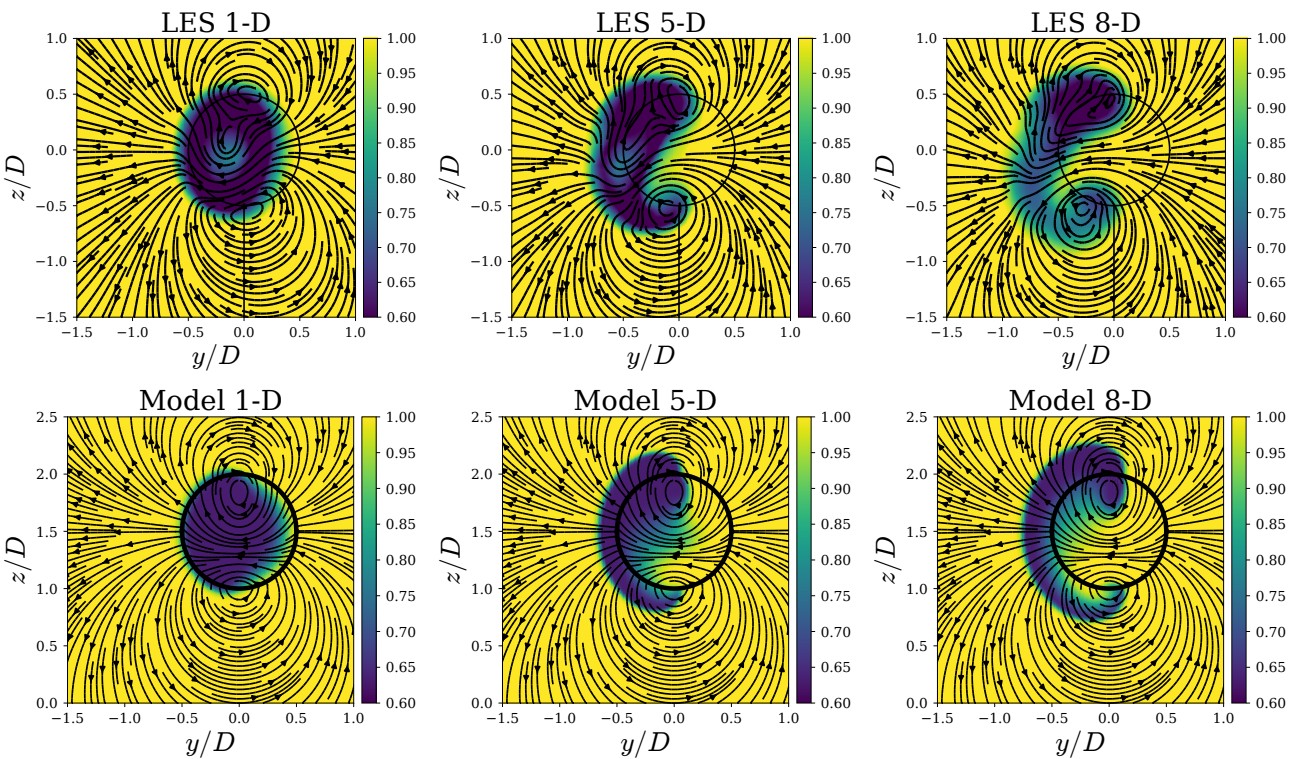

**Figure 3.** Comparison of streamwise velocity contours between a large-eddy simulation using an actuator line model under uniform inflow from Howland et al. (2016) (top) and the proposed model (bottom).The streamlines show the spanwise velocity components.

Figure 4 shows the axial velocity along a horizontal line at hub height for different downstream locations from the simulations in Figures 2 and 3. There is good agreement between the simulations and the model (in general), although some differences can be observed. Near the edges of the wake there is an acceleration in the LES, which is caused by the blockage effect, and the model is not able to capture this. In the case of the ALM, the main difference can be attributed to the different initial condition. The model assumes a step function, which is different from the wake resolved by the LES. However, the general shape of the wake and its deviation to the sides, are well-captured by the model.

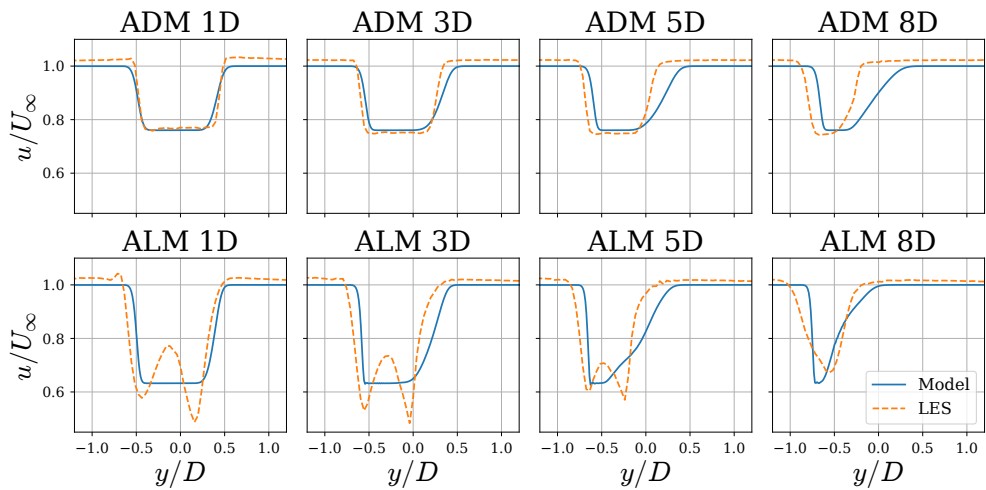

**Figure 4.** Comparison of axial velocity along a horizontal line between a large-eddy simulation using an actuator disk model (top) and actuator line model (bottom) under uniform inflow from Howland et al. (2016) and the proposed model.

Figure 5 shows the axial velocity along a vertical line that passes through the center of the rotor for different downstream locations from the simulations in Figures 2 and 3. Good agreement can be observed from the profiles between the model and LES. The general behavior of the profiles is well captured by the model. In the case of the ADM, differences in the near wake are present due to the inclusion of the tower model. The profiles from the model have sharper gradients near the edges, showing that a turbulence model should be present to incorporate the diffusion due to turbulent mixing.

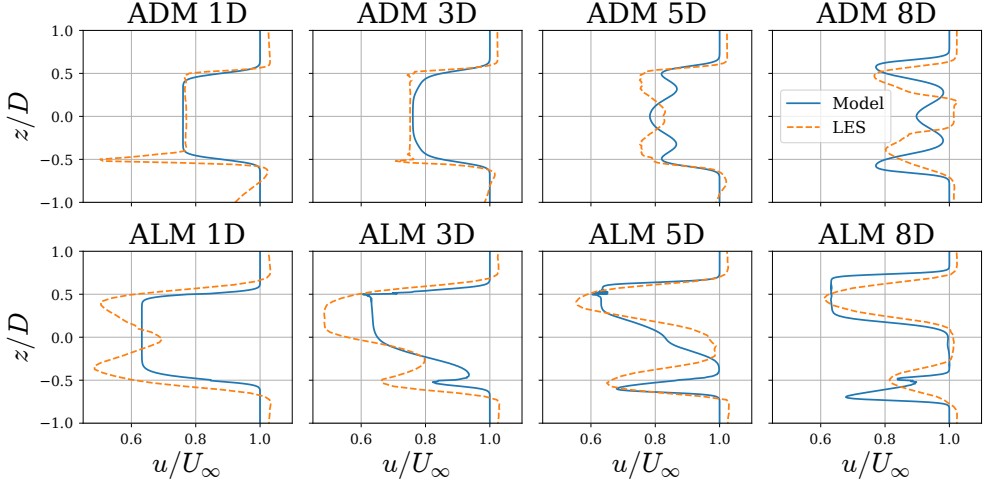

**Figure 5.** Comparison of axial velocity along a vertical line passing through the center of the rotor between a large-eddy simulation using an actuator disk model (top) and actuator line model (bottom) under uniform inflow from Howland et al. (2016) and the proposed model.

## 4.2   Large-Eddy Simulations using the Actuator Line Model in the Atmospheric Boundary Layer

The framework presented can easily be extended by adding more features. As an example, we present a comparison of the model with an LES of a wind turbine inside a neutral atmospheric boundary layer with a yaw angle, $\gamma = 20^o$. The large eddy simulation was performed using the Simulator fOr Wind Farm Applications (SOWFA) from the National Renewable Energy Laboratory (Churchfield and Lee, 2012). We also include the Gaussian wake model from Bastankhah and Porté-Agel (2016) in the comparison.

To add the effects of the atmosphere to the curled wake model, a vertical profile of velocity in the streamwise direction is added to the base solution. Also, a linear spanwise velocity component is added to the base solution to take veer into account, although this had little effect on the results presented. The veer profile was chosen as a linear profile that matched the inflow from the LES results. Figure 6 shows the inflow wind vertical profile 3 diameters upstream of the rotor for the LES and the one specified from the model, and the resulting turbulent viscosity as a function of height. A power law with a shear exponent of $\alpha = 0.15$ was chosen in the model to match the LES inflow condition. The turbulence intensity at hub height from the LES is 10%.

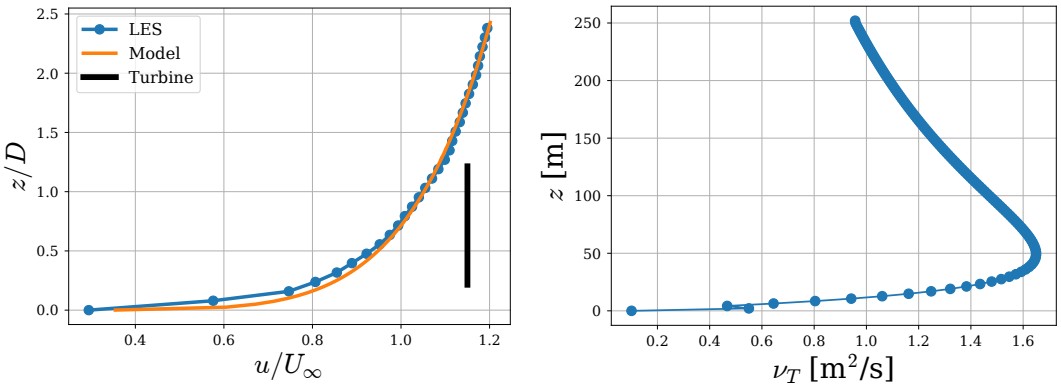

**Figure 6.** Atmospheric boundary layer inflow (left) from the LES and from a power law used in the model with shear exponent $\alpha = 0.15$. The corresponding turbulence viscosity profile (right) as a function of height is shown.

Figure 7 shows the mean streamwise velocity contours for the LES of a wind turbine inside a neutral atmospheric boundary layer and results from the model. In general, there is good agreement between the model and the simulation. We can see that the main difference comes from the wake in the LES diffusing more than in the model. This is expected because the turbulence model does not take into account the turbulence generated by the turbine wake, only the turbulence caused by the velocity gradients in the atmospheric boundary layer. There are also differences resulting from the lateral motion of the vortices. The present model does not take into account the convection of the vortices. This is shown in Figure 7, where the top and bottom vortices stay at the same place when using the model. In reality, these vortices are convected to the sides, as shown in the LES.

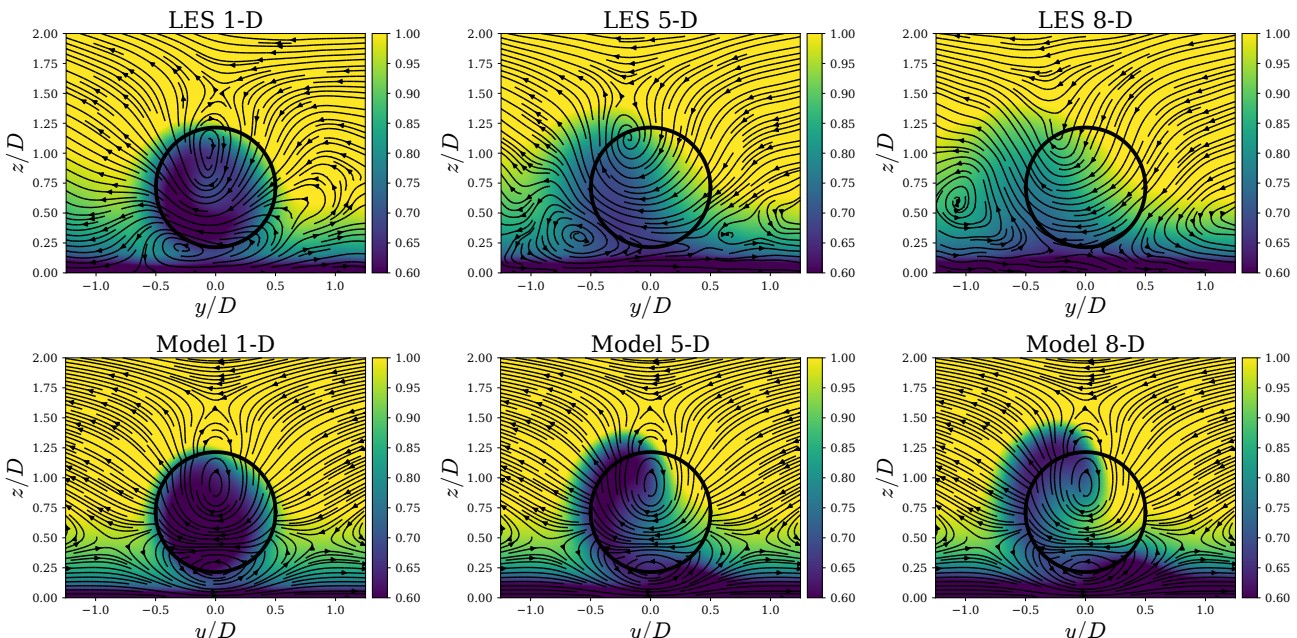

**Figure 7.** Comparison of streamwise velocity contours for the proposed model (bottom) with large-eddy simulations of a wind turbine inside the atmospheric boundary layer (top). The streamlines show the spanwise velocity components.

Figure 8 presents velocity along a horizontal and vertical lines passing through the center of the rotor comparing the model presented, the Gaussian wake model from Bastankhah and Porté-Agel (2016) and large-eddy simulations. The Gaussian model follows the implementation by Bastankhah and Porté-Agel (2016) with the wake growth rate of $k_x = k_y = 0.022$. There are differences present in the proposed model, which we attribute to the simplifications of the proposed model and the turbulence model. However, there is good agreement in in terms of near wake predictions. From the vertical profiles, we can see that further downstream the agreement between the curled wake model and the LES deteriorates. This is because the turbulent diffusion from the model does not provide enough dissipation, and because the vortices do not decay nor are they convected to the side. This means that the vortices will convect the wake deficit providing unrealistic stronger wake deficits further downstream.

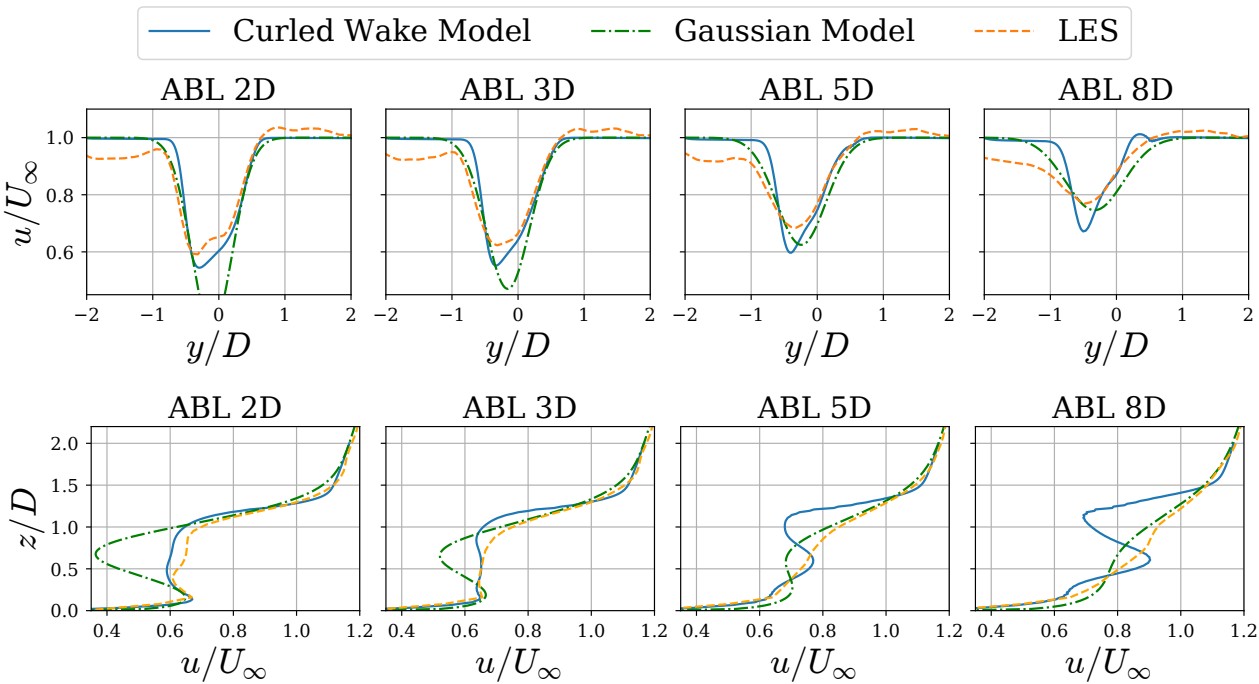

**Figure 8.** Plots across a horizontal (top) and vertical (bottom) lines passing through the center of the rotor for a large-eddy simulation using a neutral atmospheric boundary layer simulation of a turbine in $20^o$ yaw using an actuator line model and the curled wake model.

In the LES and proposed model, the curled wake shape is produced in the near wake. As the wake evolves, the turbulence diffuses the curled wake shape, and eventually, it becomes more similar to a Gaussian wake, as observed by Bastankhah and Porté-Agel (2016). In contrast to the LES and curled wake model, the Gaussian model predicts a very symmetric wake at every downstream location and is not able to predict the complex shape of the curled wake. Further downstream, a self-similar

profile seems to be a good description of the wake, since the turbulence has diffused most of the structures caused by the curling mechanism.

It is difficult to track a wake centerline in the curled wake model and the LES. The curled wake is characterized by a complex three-dimensional structure and a wake center is not really descriptive of this mechanism, especially in the near wake. Figure 9 shows lateral displacement of the center of the wake for LES, the curled wake model, the Gaussian model and the model

from Shapiro et al. (2018). The curled wake and LES cases compute this displacement by averaging a collection of tracers around the center of the rotor inside a radius of $0.2D$. We see that all the models agree relatively well in predicting the lateral displacement. We argue that the displacement is not a proper measure of the wake because it cannot track the non-symmetric and more complex shape of the curled wake. The tracer is not only displaced laterally, but also in the vertical direction. To properly represent the curled wake and its displacement, a more robust and three dimensional model, such as LES and/or the

curled wake model, should be used.

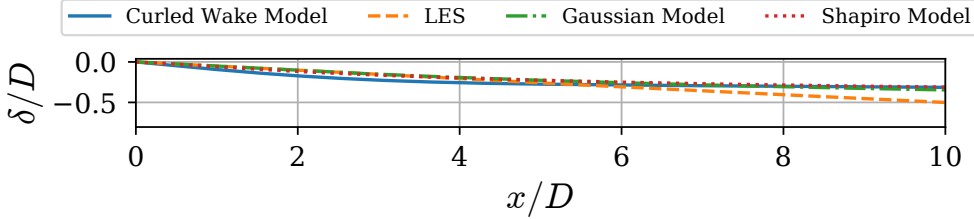

**Figure 9.** Wake displacement $\delta$ non-dimensionalized by rotor diameter $d$ along the spanwise coordinate as a function of non-dimensional downstream location $x/D$.

## 5    Controls-Oriented Modeling

We now test the proposed model inside the FLORIS framework (Gebraad et al., 2016; Annoni et al., 2018). We compare the new curled wake model to the two-dimensional Gaussian steering model from Bastankhah and Porté-Agel (2016). In the Gaussian steering model (Bastankhah and Porté-Agel, 2016), the wake deflection due to yaw misalignment of turbines is

5    defined by doing budget analysis on the Reynolds- averaged Navier-Stokes equations. In the curled wake model, the wake steering is computed by solving a linearized version of the Navier-Stokes equations.

First, we run a case of two turbines aligned with the flow and the upstream turbine is yawed by $25^o$. Figure 10 shows the streamwise velocity profiles for a FLORIS simulation with the curled wake model and the Gaussian model from the Bastankhah and Porté-Agel (2016) model. We observe that in the curled wake model, the wake of the second turbine is affected by the

10    curl of the first turbine. It was observed by Fleming et al. (2017), that the curled wake mechanism does affect the wake of the second turbine, but current yaw steering models are not able to take this effect into account. This is one of the main advantages of the curled wake model, because secondary wake steering has recently been observed and can be used to optimize wind farm performance (Fleming et al., 2017).

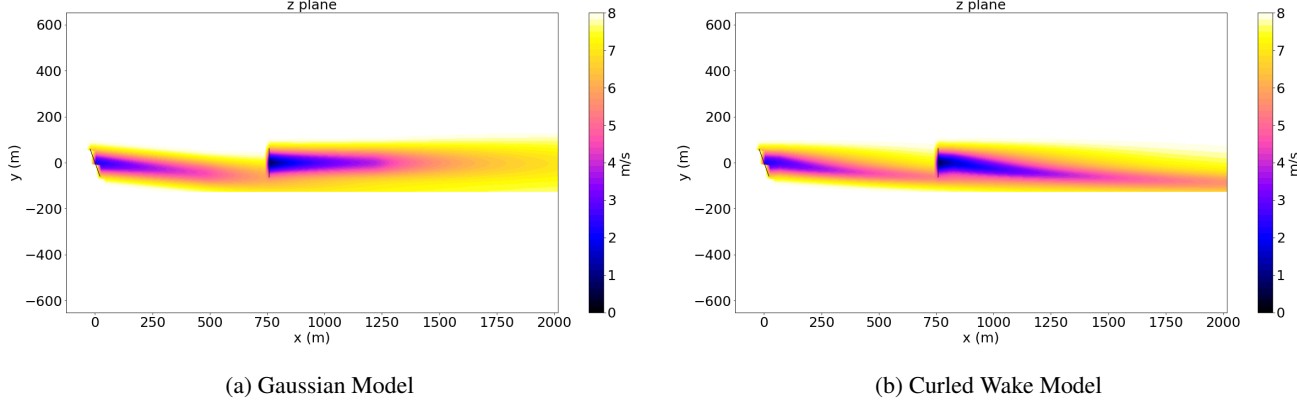

(a) Gaussian Model                        (b) Curled Wake Model

**Figure 10.** FLORIS simulation of two aligned wind turbines yawing the first turbine $25^o$.

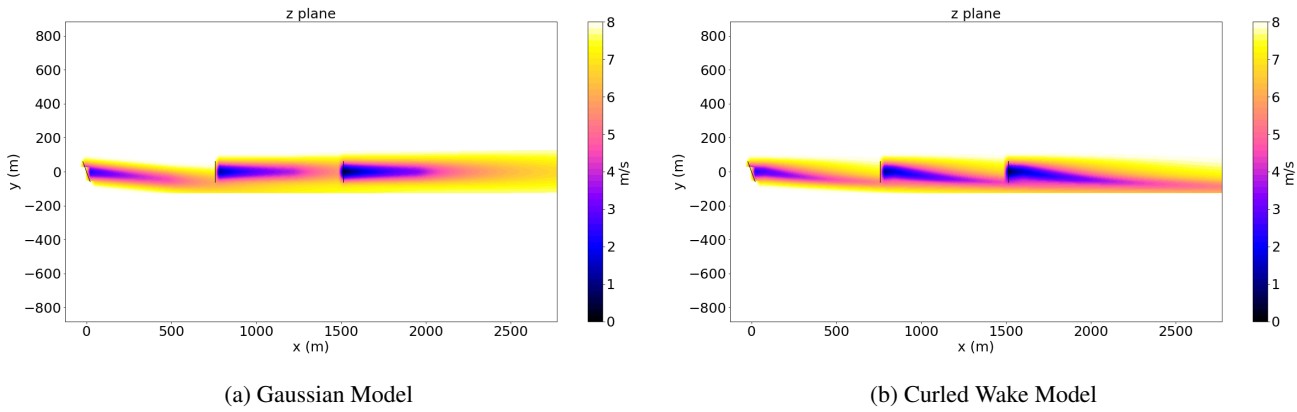

(a) Gaussian Model                (b) Curled Wake Model

**Figure 11.** FLORIS simulation of three aligned wind turbines yawing the first turbine $25^o$.

Now, we present results for three turbines aligned with the upstream turbine yawed $25^o$. The curled wake model predicts deflections up to the third turbine's wake. This approach addresses previous concerns about models not being able to capture wake deflection from downstream turbines (Fleming et al., 2017).

Table 1 shows a comparison between the power predictions and performance of the Gaussian and curled wake models and simulations performed in SOWFA. In the case of 2 turbines, both models agree very well in terms of power predictions. However, we notice that the power predictions from the curled wake over-predict results from LES in the case of 3 aligned turbines. This outcome is expected because the vortices resulting from curl do not decay as they travel downstream. Without a decay model, the spanwise velocities from the yawed turbine, would never decay. In reality, these vortices decay due to the turbulence in the atmosphere and in the wake.

The curled wake model provides improvements in predicting power gains for more than two turbines in a row. This outcome is because the vortices from the first turbine are propagated downstream. However, because the vortices do not decay in time, the power may be overpredicted.

| Model | Gaussian | Curl | SOWFA |
|---|---|---|---|
| Two-turbine power gain | 4.2% | 4.3% | 5.3% |
| Three-turbine power gain | 4.9% | 13.4% | 9.2% |
| Run time | 0.05 s | 0.5 s | 2 days |

**Table 1.** Power percentage improvements for the case with and without steering for the Gaussian model, curled wake model, and SOWFA.

## 6 Possible Improvements for the Model

The key differences between the model and simulations can be summarized as follows:

1. The vortices caused by the curl effect in the model do not change their position and do not decay. In reality, these vortices induce motion on each other and are advected by the free-stream flow, which may have a lateral component.

2. The turbulence model does not take into account the wind turbine wake. It can only take into account the turbulence from the atmospheric boundary layer background flow. This is why the wake decays faster in the large-eddy simulations compared to the model.

3. The vortices in the model do not decay with downstream distance. In reality, vortices decay because of the radial diffusion of tangential momentum.

4. The model does not take into account all the nonlinear interactions present in the simulation. For this reason, the model is only able to capture the behavior of the larger scales, and hence, not all the details of the flow (such as the deformation of the vortices) can be captured.

The model can be further improved by taking some of these factors into account. However, the present model is able to capture the main dynamics of the curled wake with a reduced computational cost. Further improvements are part of future work.

## 7 Conclusions

A new model has been proposed to study the aerodynamics of the curled wake. The model solves a linearized version of the Navier-Stokes momentum equation with the curl effect added as a collection of vortices with an elliptic distribution shed from the rotor plane. The main difference between the model presented and the Gaussian models from Bastankhah and Porté-Agel (2016) and Shapiro et al. (2018) is that there is no assumption on the shape of the wake, apart from its initial condition. The wake profile generated by the curled wake model is the solution to the linearized momentum equation with some assumptions. This allows the wake to take any shape, which in many cases was observed to differ significantly from a Gaussian wake.

The model has the ability to include several features of the wake including effects due to yaw ('curl'), wake rotation, a boundary layer profile, and turbulence modeling. The model has been implemented and tested to reproduce curled wake profiles. Good agreement is observed when comparing the model to large-eddy simulations of flow past a yawed turbine using an actuator disk/line model. The model was implemented and tested using the FLORIS framework. Good agreement was observed in predicting power extraction by yawing the first turbine in a row of two and three turbines. We observe that the effects of the vortices shed by a yawed turbine propagate for downstream distances longer than the separation between two turbines. This means that a yawed turbine can be used to redirect, not only its own wake, but the wake of other downstream turbines as well. Also, we note that the shed vortices allow for spanwise velocity components, which are vital when considering wake redirection and wind farm controls. The vortices generated are not limited to only yawing, as they can also be used for tilt and combinations of tilt and yaw. This work sets a foundation for a simplified wake steering model to be used in a more general wind farm control-oriented framework. Future work consists of improving the curled wake model with emphasis on implementing a robust decay model for the vortices and comparing the model to experimental data.

*Competing interests.* The authors declare that they have no conflict of interest.

*Acknowledgements.* The authors would like to acknowledge Charles Meneveau and Patrick Hawbecker for suggestions on turbulent modeling in the atmospheric boundary layer and Patrick Moriarty for providing wind turbine aerodynamics insights. Simulations for the NREL code SOWFA were performed using the National Renewable Energy Laboratory's Peregrine high-performance computing system. Numerical
5   implementation and plots were done using python with numpy, maptlotlib, and scipy libraries.

    This work was authored by the National Renewable Energy Laboratory, operated by the Alliance for Sustainable Energy, LLC, for the U.S. Department of Energy (DOE) under Contract No. DE-AC36-08GO28308. Funding provided by the U.S. Department of Energy Office of Energy Efficiency and Renewable Energy Wind Energy Technologies Office. The views expressed in the article do not necessarily represent the views of the DOE or the U.S. Government. The U.S. Government retains and the publisher, by accepting the article for publication,
10   acknowledges that the U.S. Government retains a nonexclusive, paid-up, irrevocable, worldwide license to publish or reproduce the published form of this work, or allow others to do so, for U.S. Government purposes.

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
