# Peer review of "The Aerodynamics of the Curled Wake: A Simplified Model in View of Flow Control"

_Wind Energy Science, 2018_

## Referee Comment (RC1) · Anonymous Referee #1 · 24 Aug 2018

I consider this to be a well-written and well-structured paper with a clear contribution. I have to say that I am not an aerodynamicist and therefore cannot asses every detail of this paper. I have some minor comments/questions/suggestions:

- It is not clear if this model replaces or extends the FLORIS model as defined in (Gebraad). The caption of Figure 9 states that they use the FLORIS model but with different features (Gaussian vs Curl)? It would be good to explain how the FLORIS model is extended with the Curled Wake model or the Gaussian model.

- On pg 11, "We observe that both models agree very well in terms of power production" How do we see that? Do we see that in Table 1? What is the added value of the 2 turbine simulation? The effects highlighted in the 2 turbine simulations should also be visible in the three turbine case.

- Table 1, 3 turbine case. It seems that the Curled model is as bad as the Gaussian model (if you compare with SOWFA). The authors argue that this has to do with the fact that there is no decay model. Adding a decay model would also influence the two turbine case. It would be good to add some kind of tuned decay model to really show the strength of the model.

- In general, it is not clear how the different engineering models are tuned.

- Conclusions, the authors state that this work sets a foundation for a simplified wake steering model in a more wind farm control-oriented framework. I believe the authors are referring to an estimation step followed by an optimization step (they can make this more explicit). The authors also increase the complexity of the model which typically means more tuning variable and thus a more complex control problem. It would be good to add a discussion on how this model can be used with respect to the state-of-the-art.

- A literature overview with containing other control-oriented models is lacking.

- In figure 3 and 4 it is not specified which quantity is plotted

- I believe it is "an LES" instead of "a LES"

---

## Referee Comment (RC2) · Anonymous Referee #2 · 5 Oct 2018

In the manuscript, the authors present a model for the curled wake based on approximations to the Navier-Stokes equations. They compared the model predictions with the LES data of wind turbine wakes in uniform and turbulent inflow. The topic is interesting and useful for the wind-energy community, and the manuscript is well-written. However, there are some issues which are required to be addressed.

1. The results obtained from the other models, in particular, the ones proposed by Shapiro et al. (2018) and Bastankhah and Porte-Agel (2016), could be added to the text (in Fig. 5 and 7). In the current format, it is not possible to compare the performance of the proposed model with the other ones.

2. Since the experimental data of wind turbine wakes in yawed conditions is available (e.g., Bastankhah and Porte-Agel, 2016), it would be more useful if the model is also

compared with the experimental data.

3. In addition to Fig. 5 and 7, the vertical profiles of the wind velocity should be added to the manuscript to better compare the model with the simulations.

4. A figure showing the lateral displacement of the wake with downwind distance could be added to the text, and it should be compared with the other existing models.

5. In Table 1 and the related text, it is not mentioned which method for the wake superposition is used in the Gaussian wake model (e.g., Katic, Lissaman, Voutsinas, or a different one). Please clarify this issue in the text.

6. In Fig. 4 and 6, why the predictions from the proposed model is different in the laminar inflow for the ADM and ALM? Is there any difference in the simulation setup using different turbine models? Can the model differentiate between the ADM and ALM?

7. Regarding the eddy viscosity model, there is no wall in the simulation under laminar inflow condition. How the eddy viscosity is computed in that case? Is it zero? For the turbulent case, it would be useful if the authors could show the comparison between the eddy viscosity model in equation (15) and the LES.

8. It is not clear how the model includes the turbulence level in the incoming flow. Is it included in the model though the effective viscosity?

9. It is suggested that, in the ABL case, the authors consider another yaw angle (e.g. 30o) to make the model validation more complete.

10. In the ABL case, it would be useful if the incoming wind characteristics (i.e., vertical profiles of the mean wind velocity (in the log scale) and the turbulence intensity) are added to the text.

---

## Author Comment (AC1) · 5 Nov 2018

Reviewer 1

I consider this to be a well-written and well-structured paper with a clear contribution. I have to say that I am not an aerodynamicist and therefore cannot asses every detail of this paper. I have some minor comments/questions/suggestions:

The authors appreciate the positive feedback and useful comments from the reviewer. The comments have been addressed below. All changes are marked in blue in the manuscript.

- It is not clear if this model replaces or extends the FLORIS model as defined in (Gebraad). The caption of Figure 9 states that they use the FLORIS model but with different features (Gaussian vs Curl)? It would be good to explain how the FLORIS model is extended with the Curled Wake model or the Gaussian model.

**Response:**

The model presented in this paper extends the use of the Floris frame-work. This is a wake model which is a new option inside of Floris. This explanation has been incorporated to the text:

"The model proposed in this work is a new wake model and it has been incorporated to the FLORIS framework. The model is used to compute the wake from each turbine in a wind farm. After computing the individual wakes, they are added using the sum of squares method (Katic et al., 1987)."

- On pg 11, "We observe that both models agree very well in terms of power production" How do we see that? Do we see that in Table 1? What is the added value of the 2 turbine simulation? The effects highlighted in the 2 turbine simulations should also be visible in the three turbine case.

**Response:**

We agree with the reviewer and the results presented in this section were unclear. This section has been edited as follows:

"Table 1 shows a comparison between the power predictions and performance of the Gaussian and curled wake models and simulations performed in SOWFA. 5 In the case of 2 turbines, both models agree very well in terms of power predictions. However, we notice that the power predictions from the curled wake over-predict results from LES in the case of 3 aligned turbines. This outcome is expected because the vortices resulting from curl do not decay as they travel downstream. Without a decay model, the spanwise velocities from the yawed turbine, would never decay. In reality, these vortices decay due to the turbulence in the atmosphere and in the wake."

- Table 1, 3 turbine case. It seems that the Curled model is as bad as the Gaussian model (if you compare with SOWFA). The authors argue that this has to do with the fact that there is no decay model. Adding a decay model would also influence the two turbine case. It would be good to add some kind of tuned decay model to really show the strength of the model.

**Response:**
Yes, we agree that the decay model is a much-needed capability for the model. This is part of ongoing/future work. A statement has been added to the conclusion to reassure this:

"Future work consists of improving the curled wake model with emphasis on implementing a robust decay model for the vortices and comparing the model to experimental data."

- In general, it is not clear how the different engineering models are tuned.
**Response:**
There is minimum tuning of the parameters in the curled wake model. Since the curled wake model solves a simplified version of the Navier-Stokes equations, most of the effects can be incorporated through physics models rather than tunable parameters. All of the tunable parameters included in the curled wake model are explained in the manuscript.

- Conclusions, the authors state that this work sets a foundation for a simplified wake steering model in a more wind farm control-oriented framework. I believe the authors are referring to an estimation step followed by an optimization step (they can make this more explicit). The authors also increase the complexity of the model which typically means more tuning variable and thus a more complex control problem. It would be good to add a discussion on how this model can be used with respect to the state-of-the-art.
**Response:**
The model presented in the manuscript is only a wake model. This wake model can be incorporated into other optimization routines in a control-oriented framework such as Floris. We agree that the module is more computationally expensive than other models, but this is necessary to capture the curl physics. This model is a new option (under development) in the Floris frame-work.

- A literature overview with containing other control-oriented models is lacking.
**Response:**
An overview of Floris, and some of the wake models used has been added. A reference to a more complete review on wake models has also been added in the introduction:

"FLOw Redirection and Induction in Steady State (FLORIS) is a software framework used for wind plant performance optimization (Gebraad et al., 2016; Annoni et al., 2018). A wake model is used in FLORIS to compute the effect of wind turbine wakes on downstream turbines. Different models can be used inside of FLORIS to compute the wind turbine wakes, including the Jensen and Gaussian wake models (Bastankhah and Porté-Agel, 2016; Jensen, 1983). A review of control-oriented models can be found in Annoni et al. (2018)."

- In figure 3 and 4 it is not specified which quantity is plotted
**Response:**

The caption in the figures has been changed to:

"Comparison of streamwise velocity contours between a large-eddy simulation using an actuator line model under uniform inflow from \citet{howland2016} (top) and the proposed model (bottom). The streamlines show the spanwise velocity components."

- I believe it is "an LES" instead of "a LES"
**Response:**
Yes, this has been changed accordingly.

---

## Author Comment (AC2) · 6 Nov 2018

Reviewer 2

In the manuscript, the authors present a model for the curled wake based on approximations to the Navier-Stokes equations. They compared the model predictions with the LES data of wind turbine wakes in uniform and turbulent inflow. The topic is interesting and useful for the wind-energy community, and the manuscript is well-written. However, there are some issues which are required to be addressed.

The authors appreciate the positive feedback from the referee. The comments have been addressed below. The responses are marked in blue in the manuscript.

1. The results obtained from the other models, in particular, the ones proposed by Shapiro et al. (2018) and Bastankhah and Porte-Agel (2016), could be added to the text (in Fig. 5 and 7). In the current format, it is not possible to compare the performance of the proposed model with the other ones.

**Response:**

We agree with the reviewer and the profiles suggested for the Gaussian model have been added. We noticed that in the cases with uniform inflow, it is difficult to compare the curled wake model to the Gaussian model. In these conditions the Gaussian model is only able to predict the wake profiles too far downstream. However, in the case of the atmospheric boundary layer, the Gaussian model works well. Section 4.2 has been re-written by adding the results from the Gaussian model.

2. Since the experimental data of wind turbine wakes in yawed conditions is available (e.g., Bastankhah and Porte-Agel, 2016), it would be more useful if the model is also compared with the experimental data.

**Response:**

We agree with the reviewer that comparisons to experiments are valuable. However, in order to keep the paper concise and focused on the derivation of the model and comparisons to LES, we will leave this suggestion as part of future work. This has been addressed in the conclusions:

"Future work consists of improving the curled wake model with emphasis on implementing a robust decay model for the vortices and comparing the model against experimental data."

3. In addition to Fig. 5 and 7, the vertical profiles of the wind velocity should be added to the manuscript to better compare the model with the simulations.

**Response:**

We agree that these figures would help to better compare the model and the LES. Both these figures have been added and the following text has been added to the manuscript:

"Figure 5 shows the axial velocity along a vertical that passes through the center of the rotor for different downstream locations from the simulations in Figures 2 and 3. Good agreement can be observed from the profiles between the model and LES. The general behavior of the profiles is well capture by the model. In the case of the ADM, differences in the near wake are present

due to the inclusion of the tower model. The profiles from the model have sharper gradients near the edges, showing that a turbulence model should be present to incorporate the diffusion due to turbulent mixing."

"Figure 8 presents velocity along a horizontal and vertical lines passing through the center of the rotor comparing the model presented, the Gaussian wake model from Bastankhah and Porté-Agel (2016) and large-eddy simulations. There are differences present in the proposed model, which we attribute to the simplifications of the proposed model and the turbulence model. However, there is good agreement in the curled wake model in terms of near wake predictions. As the wake moves downstream, the Gaussian model seems to better capture the general shape, but the location of the wake seems too far off in the negative y direction. From the vertical profiles, we can see that further downstream the agreement between the curled wake model and the LES deteriorates. This is because the turbulent diffusion from the model does not provide enough dissipation, and because the vortices do not decay nor are they convected to the side. This means that the vortices will convect the wake deficit providing unrealistic stronger wake deficits further downstream."

4. A figure showing the lateral displacement of the wake with downwind distance could be added to the text, and it should be compared with the other existing models.
**Response:**
We have added the figure with the following discussion:

"It is difficult to track a wake centerline in the curled wake model and the LES. The curled wake is characterized by a complex three-dimensional structure and a wake center is not really descriptive of this mechanism, specially in the near wake. Figure 9 shows lateral displacement of the center of the wake. The curled wake and LES cases compute this displacement by averaging a collection of tracers around the center of the rotor inside a radius of 0:2D. It is interesting to note that both models underpredict the displacement in the far wake compared to the LES. We argue that the displacement is not a proper measure because it cannot track the non-symmetric and more complex shape of the curled wake. The tracer is not only displaced laterally, but also in the vertical direction. To properly represent the curled wake and its displacement, a more robust and three dimensional model, such as LES and/or the curled wake model, should be used."

5. In Table 1 and the related text, it is not mentioned which method for the wake superposition is used in the Gaussian wake model (e.g., Katic, Lissaman, Voutsinas, or a different one). Please clarify this issue in the text.
**Response:**
The sum of squares method has been used to superpose the wakes. This has been added to the introduction:
"After computing the individual wakes, they are added using the sum of squares method (Katic et al., 1987)."

6. In Fig. 4 and 6, why the predictions from the proposed model is different in the laminar inflow for the ADM and ALM? Is there any difference in the simulation setup using different turbine models? Can the model differentiate between the ADM and ALM?

**Response:**

The difference between ADM and ALM in the model is the addition of rotation. This has been explained in the text:

"The difference between the model implementation in this case, and the case of the actuator disk model is that rotation of the wake has been added"

7. Regarding the eddy viscosity model, there is no wall in the simulation under laminar inflow condition. How the eddy viscosity is computed in that case? Is it zero? For the turbulent case, it would be useful if the authors could show the comparison between the eddy viscosity model in equation (15) and the LES.

**Response:**

In the case of uniform inflow, the turbulence model is not present. However, to stabilize the numerical solution a small viscous term has to be added. This has been explained in the manuscript:

"In the case of laminar inflow, an arbitrary viscosity needs to be added to stabilize the numerical solution. After experimenting with different values, an effective viscosity based on a Reynolds number Re = 10^4 proved to be sufficient to stabilize the solution."

8. It is not clear how the model includes the turbulence level in the incoming flow. Is it included in the model though the effective viscosity?

**Response:**

Yes, the turbulent viscosity is specified in terms of the atmospheric boundary layer profile. There is no direct dependency on turbulence intensity. The current model is only valid for neutral stability and improvements to the turbulence model which take into account stability is part of future work.

9. It is suggested that, in the ABL case, the authors consider another yaw angle (e.g. 30o) to make the model validation more complete.

**Response:**

Results in the manuscript are presented for a range of yaw angles between 20 and 30 degrees. We have notices similar behavior of the model for smaller angles but have focused on these angles because they show best the curled wake mechanism and its effect on the wake.

10. In the ABL case, it would be useful if the incoming wind characteristics (i.e., vertical profiles of the mean wind velocity (in the log scale) and the turbulence intensity) are added to the text.

**Response:**
We agree with the reviewer. We have added the inflow profiles and a description:

"Figure 6 shows the inflow wind vertical profile 3 diameters upstream of the rotor for the LES and the one specified from the model, and the resulting turbulent viscosity as a function of height. A power law with a shear exponent of alpha = 0.15 was chosen in the model to match the LES inflow condition."

---

## Referee Report (RR1)

Review of Manuscript # wes-2018-57

"The Aerodynamics of the Curled Wake: A Simplified Model in View of Flow Control"

Luis A. Martínez-Tossas et al.

In the revised manuscript, the authors addressed the main issues raised in the first review. Hence, I would like to accept the manuscript. However, there is a minor issue that can be easily addressed by the authors:

I believe that the difference between the Gaussian model and the simulation results shown in Table 1 is related to the superposition method used in this paper. It is already well known that the Katic superposition overestimates the power loss especially for the aligned configurations. As already shown in several studies, the reference velocity in the energy-deficit superposition could not be $u_0$ as proposed by Katic and used in the current manuscript. In particular, the superposition should be written as: $U_i = u_0 - [\sum_k (u_k - u_{ki})^2]^{1/2}$ (Voutsinas et al. 1990) instead of $U_i = u_0 - [\sum_k (u_0 - u_{ki})^2]^{1/2}$ (Katic et al. 1987). Otherwise, the analytical model overestimates the power loss for the downstream wind turbines. I suggest that the authors use the Voutsinas superposition ($U_i = u_0 - [\sum_k (u_k - u_{ki})^2]^{1/2}$) and update the results shown in Table 1. I think the difference between the model and the simulation results would decrease by this modification.

Reference. Voutsinas, S.; Rados, K.; Zervos, A. On the analysis of wake effects in wind parks. Wind Eng. 1990, 14, 204–219.

---

## Author Response (AR2)

Review of Manuscript # wes-2018-57
"The Aerodynamics of the Curled Wake: A Simplified Model in View of Flow Control"
Luis A. Martínez-Tossas et al.

In the revised manuscript, the authors addressed the main issues raised in the first review. Hence, I would like to accept the manuscript. However, there is a minor issue that can be easily addressed by the authors:

We appreciate the positive feedback from the reviewer. The response to the comments are marked in blue.

I believe that the difference between the Gaussian model and the simulation results shown in Table 1 is related to the superposition method used in this paper. It is already well known that the Katic superposition overestimates the power loss especially for the aligned configurations. As already shown in several studies, the reference velocity in the energy-deficit superposition could not be $u0$ as proposed by Katic and used in the current manuscript. In particular, the superposition should be written as: $Ui=u0-[\Sigma(uk-uki)2\ k]1/2$ (Voutsinas et al. 1990) instead of $Ui=u0-[\Sigma(u0-uki)2\ k]1/2$ (Katic et al. 1987). Otherwise, the analytical model overestimates the power loss for the downstream wind turbines. I suggest that the authors use the Voutsinas superposition ($Ui=u0-[\Sigma(uk-uki)2\ k]1/2$) and update the results shown in Table 1. I think the difference between the model and the simulation results would decrease by this modification.

Reference. Voutsinas, S.; Rados, K.; Zervos, A. On the analysis of wake effects in wind parks. Wind Eng. 1990, 14, 204–219.

Yes, we agree that the different superposition methods can cause differences in the results. We tried running the superposition method suggested and still found similar results. The difference in power gains between the different superposition methods was less than 1%. The changes observed from the different superposition methods are still small compared to the improvements obtained from the curled wake model. We believe that the model presented adds important physics to the model that cannot be captured with the other models. We have seen that these improvements are more significant than using different superposition methods.

I have joined the review process from this round, so my comments will be based on the latest version of the manuscript as well as previous reviewers' comments. In this manuscript, a model based on linearized form of Navier-Stokes equations is developed that can capture and predict the curled wakes of yawed turbines. The paper is overall clear, well written, and it contains results that are indeed interesting and useful for the wind energy community, especially for those interested in yaw angle control. I especially appreciate the part where model advantages as well as its limitations are clearly presented. However, I have some major concerns over the way results are presented in this manuscript. I therefore believe the paper will merit publication in the wind energy science (WES), provided the authors can address my comments:

We thank the reviewer for their positive and detailed feedback. The responses to the comments/suggestions are marked in blue.

• It is useful that the model results are compared with those of the Gaussian model. However, the paper still suffers from the lack of quantitative comparison with the more conceptually similar model proposed by Shapiro et al. (2018). In my opinion, it is quite important to explicitly clarify the difference of the present work with the mentioned study in terms of model derivation, and then also add predictions of Shapiro et al. model to Figs. 8 and 9. I believe this comparison can be informative and useful for readers.

There are some key differences in the model from Shapiro et al and the one presented. The model from Shapiro et al is more conceptually similar to the Gaussian wake model from Bastankhah. The model from Shapiro et al assumes a Gaussian profile of the wake. The model we present makes no assumptions on the wake profile. It solves a linearized version of the momentum equation to obtain the wake profile. As shown throughout the paper, this shape cannot always be characterized by a Gaussian profile. There is one key aspect that both models have in common: the elliptic distribution of vorticity. We have referenced the Shapiro paper when we mention the elliptical distribution. The differences between the models has been added to the conclusions:

"The main difference between the model presented and the Gaussian models from Bastankhah and Porté-Agel (2016) and Shapiro et al. (2018) is that there is no assumption on the shape of the wake, apart from its initial condition. The wake profile generated by the curled wake model is the solution to the linearized momentum equation with some assumptions. This allows the wake to take any shape, which in many cases was observed to differ significantly from a Gaussian wake."

We have also included the results from Shapiro et al in Figure 9 and added it to the text:

"Figure 9 shows lateral displacement of the center of the wake for LES, the curled wake model, the Gaussian model and the model from Shapiro et al. (2018)."

• Eq. 4: u'*du'/dx is neglected although it is later considered in Eq. 16.

This is a great observation. Yes, this term has been included in the numerical solution. However, we have noted little difference if it is included or not. This term can be included in the current form of the equation because of its explicit nature. In the case of an implicit algorithm, it would be more difficult to include this term. The following explanation has been added to the text:

"Because of the explicit form of the equation, it is possible to include the non-linear term $u' \frac{\partial u'}{\partial x}$, which is not present in Equation 4. We note that this term has a small effect on the solution, and not including it provides similar results."

• Figure 6: I understand that incoming turbulence is not directly used in the proposed model, but I think it is of interest to report the vertical profile of turbulence intensity, or at least please report the value of the streamwise turbulence intensity at hub height.

We have added the turbulence intensity at hub height from the LES:

"The turbulence intensity at hub height from the LES is 10%."

• Page 14, line 10: "It is interesting to note that both models under-predict the displacement in the far wake compared to the LES". I found the previous sentence in contrast to what is reported in the Figure 8 for the Gaussian model predictions at x=8D.

Yes, we agree with this observation. We have eliminated this sentence from the manuscript.

• Figure 10: For the downwind turbine, which initial conditions are used for V and W? If, as mentioned in the introduction, each turbine is treated individually, and turbine wakes are added later using the sum of squares, I cannot understand how the curl of the first turbine can affect the wake of the second turbine.

The V and W components of velocity due to the curl are added linearly. This way, the wakes do see the influence from other turbines due to yaw. This has been added to the manuscript:

"In the curled wake model, the $V$ and $W$ velocity components generated by each turbine are superposed linearly."

• Figure 8: I found the near-wake deflection predicted by the Gaussian model much bigger than what I expect. For instance, the wake center deflection at x/D=3 seems to be less than 0.2D in model predictions reported by Bastankhah and Porte-Agel (2016) (See Fig. 21 in the original paper), whereas it is more than 0.5D in figure 8 of the current paper. Please report input parameters such as the wake growth rate, etc. and insure that the model is implemented properly.

Yes, we agree with this observation. We have adjusted the implementation of the Gaussian model. The figures have been updated accordingly. The input parameters used are the same as in the original paper from Bastankhah et al. This description has been added to the manuscript:

"The Gaussian model follows the implementation by Bastankhah and Porté-Agel (2016) with the wake growth rate of kx = ky= 0.022."

Minor comments:

• Introduction: I think "controls community" should be replaced by "control's community".
This has been modified to:
"The mechanism behind this effect has drawn attention, not only from fluid dynamicists because of the interesting physics phenomena happening in the wake, but also from the wind turbine controls community who intends to use it to control wind farm flows (Fleming et al., 2017)"
• Eq. 1: I suppose u, v and w are time-averaged velocity components as the effective viscosity is used. Please mention if that is the case.
Yes, this has been included in the text:
"u, v and w are the time-averaged velocity components in the streamwise, spanwise and wall-normal directions. P is the time-averaged pressure"
• Page 3, Line 3: Please explain here what U, V, W and u,v,w mean physically so that it is easier for readers to understand the assumptions made in the following.
This explanation has been included: "$U$, $V$, and $W$ represent a base flow that is responsible for convecting the wakes. And $u'$, $v'$, and $w'$ are the perturbation velocities about the base solution which represent the wake deficits."
• Eq 9: If I understood well, $z_i$ should be the location of i-th vortice with respect to the rotor coordinate system as opposed to $z_i$ defined in Eq. 7 as the one with respect to each vortex position. Please clarify this.
Great observation! This has been modified.
• Figs. 4 and 5: Please change the color for LES results as they can be hardly seen in a printed version. Add "axial" before "velocity" in the figure captions.
"Axial" has been added to the caption. The colors can be seen well when I tried to print it (both color and grayscale). This might be a problem with the printer, and not necessarily the figures.
• Page 11, line 1: Add "line" after "vertical".
This has been added.
• Page 11, line 3: Replace "capture" with "captured".
This has been changed.

[revised manuscript text omitted]